# MicroRNAs Modulate Signaling Pathways in Osteogenic Differentiation of Mesenchymal Stem Cells

**DOI:** 10.3390/ijms22052362

**Published:** 2021-02-27

**Authors:** Chiara Mazziotta, Carmen Lanzillotti, Maria Rosa Iaquinta, Francesca Taraballi, Elena Torreggiani, John Charles Rotondo, Lucia Otòn-Gonzalez, Elisa Mazzoni, Francesca Frontini, Ilaria Bononi, Monica De Mattei, Mauro Tognon, Fernanda Martini

**Affiliations:** 1Department of Medical Sciences, Section of Experimental Medicine, School of Medicine, University of Ferrara, 64b Fossato di Mortara Street, 44121 Ferrara, Italy; chiara.mazziotta@unife.it (C.M.); carmen.lanzillotti@unife.it (C.L.); mariarosa.iaquinta@unife.it (M.R.I.); elena.torreggiani@unife.it (E.T.); rtnjnc@unife.it (J.C.R.); lucia.otongonzalez@unife.it (L.O.-G.); elisa.mazzoni@unife.it (E.M.); frnfnc2@unife.it (F.F.); ilaria.bononi@unife.it (I.B.); mrf@unife.it (F.M.); 2Center for Musculoskeletal Regeneration, Houston Methodist Research Institute, 6670 Bertner Ave, Houston, TX 77030, USA; ftaraballi2@houstonmethodist.org; 3Orthopedics and Sports Medicine, Houston Methodist Hospital, 6565 Fannin Street, Houston, TX 77030, USA; 4Laboratory for Technologies of Advanced Therapies (LTTA), University of Ferrara, 70, Eliporto Street, 44121 Ferrara, Italy

**Keywords:** mesenchymal stem cells, MSCs, microRNA, miRNA, bone, osteogenic differentiation, osteogenesis, cell differentiation, stem cells, miRNAome, bone regeneration

## Abstract

Mesenchymal stem cells (MSCs) have been identified in many adult tissues and they have been closely studied in recent years, especially in view of their potential use for treating diseases and damaged tissues and organs. MSCs are capable of self-replication and differentiation into osteoblasts and are considered an important source of cells in tissue engineering for bone regeneration. Several epigenetic factors are believed to play a role in the osteogenic differentiation of MSCs, including microRNAs (miRNAs). MiRNAs are small, single-stranded, non-coding RNAs of approximately 22 nucleotides that are able to regulate cell proliferation, differentiation and apoptosis by binding the 3′ untranslated region (3′-UTR) of target mRNAs, which can be subsequently degraded or translationally silenced. MiRNAs control gene expression in osteogenic differentiation by regulating two crucial signaling cascades in osteogenesis: the transforming growth factor-beta (TGF-β)/bone morphogenic protein (BMP) and the Wingless/Int-1(Wnt)/β-catenin signaling pathways. This review provides an overview of the miRNAs involved in osteogenic differentiation and how these miRNAs could regulate the expression of target genes.

## 1. Introduction

Mesenchymal stem cells (MSCs) are multipotent cells widely studied for their potential promising properties in regenerative medicine and in clinical practice [1,2]. Isolated for the first time in 1968 from bone marrow (BM-MSCs) [3], MSCs have been obtained from several body districts and evaluated for their ability to differentiate in vitro into osteocytes, chondrocytes and adipocytes [4,5] (Figure 1).

MSCs play a key role in the natural events leading to bone repair and healing by differentiating into osteoblasts, which secrete a regenerating bone matrix. Because of their biological features, such as sustained self-renewal and expansion, ease of availability and ability to differentiate into osteoblasts, MSCs are also attractive candidates for bone tissue engineering approaches. In addition, they show anti-inflammatory and immunomodulatory effects and secrete molecules that can initiate or support tissue regeneration/replacement [5]. A large number of studies has been carried out on the biological properties of MSCs, signaling pathways involved in their differentiation and factors that might facilitate MSC osteogenic differentiation [5,6]. 

Endogenous growth factors, cytokines and signals from the extracellular matrix, mainly via integrin interactions, represent essential regulators in cell differentiation [7,8]. Further, external stimuli, such as mechanical forces or electromagnetic fields, modulate cell behavior during osteogenesis [9,10].

Biomaterials provide the biological structure that supports MSC osteogenic differentiation [2,11,12,13,14,15,16,17,18]. The addition of ions has been proved to enhance the osteogenic potential of scaffolds [19]. Biological macromolecules, such as growth factors [20] and microRNAs (miRNAs) [21], as well as biophysical stimuli [22], can also be associated with scaffolds to increase cell differentiation. 

Growing evidence shows that MSC differentiation is influenced by several epigenetic processes/factors, including histone methylation and acetylation, DNA methylation [23], non-coding RNA molecules, such as long non-coding RNAs (lncRNA) [24], and miRNAs [25,26,27,28].

MiRNAs are small, single-stranded, non-coding RNAs consisting of approximately 21–25 nucleotides (nt) and expressed in many organisms, from viruses [29,30,31] to eukaryote cells [32]. MiRNAs are able to negatively regulate gene expression at a post-transcriptional level, by interfering with target messenger RNA (mRNA) translation [33], and their functions have been widely studied in several cellular and molecular processes [34,35], including MSC osteogenic differentiation [5,6]. MiRNAs can positively or negatively regulate osteogenic differentiation by targeting transcription factors (TFs) and genes coding for either negative or positive differentiative modulators [27,36,37,38]. Different computational tools, such as TargetScan, miRanda and DIANA microT, have been developed in order to predict miRNA targets [39]. After prediction, targets have to be experimentally validated in the laboratory by PCR [40] and western blot analyses [41] to confirm the miRNA functional activity. Knowledge concerning the role of miRNAs is complex as they modulate mRNAs acting in concert with other classes of non-coding RNAs, such as long non-coding RNAs (lncRNAs) [27,42]. Further, each miRNA shows multiple activities targeting several mRNAs, thus directly or indirectly modulating the expression of a plethora of genes involved in many cellular functions [43]. 

This review is addressed to the miRNAs that, by sequence complementarity, bind mRNAs directly regulating the gene expression of molecules involved in the main signaling pathways of MSC osteogenic differentiation, such as the TGF-β/bone morphogenic protein (BMP) and the Wingless/Int-1(Wnt)/β-catenin pathway [44,45]. 

## 2. MSC Signaling Pathways in Osteogenic Differentiation

MSC osteogenic differentiation encompasses three phases: the first, also known as the proliferative phase, consists of the recruitment and expansion of osteoprogenitor cells; the second involves the cell maturation in pre-osteoblasts upon extracellular matrix (ECM) synthesis; and the third phase consists of the matrix mineralization [46].

All of these phases are modulated by several signaling pathways [47]. Among them, the TGF-β/BMP and the Wnt/β-catenin pathways [45] play critical interconnected roles and have also been closely investigated for clinical implications in bone healing [48,49,50]. TGF-β signaling appears to be mainly involved in the early phase of osteogenesis, allowing osteoprogenitor cells to differentiate into immature osteoblasts [44], whereas it inhibits osteoblast maturation, mineralization and transition into osteocytes [51,52]. Contrariwise, BMP signaling promotes almost every phase during osteoblast differentiation and maturation [44]. BMPs were the first proteins identified as inducing bone formation in vivo [53]. Among over 20 BMP proteins, BMP2, 4, 5, 6, 7 and 9 play important roles in osteogenesis [54,55]. 

TGF-βs and BMPs work through type I and type II heterodimeric kinase receptors and need Sma- and Mad-related (SMAD) proteins to transduce the signal inside the cells (Figure 2). 

Several type I and type II receptors have been identified in transducing the signal of TGF-β and BMP ligands [49,56]. The preferential type I receptor of TGF-βs is TβRI/ALK5, while the main type I receptors of BMPs are BMPR1A/ALK3, BMPR1B/ALK6 and AcvR1/ALK2. The type II kinase receptors of TGF-βs and BMPs are mainly TβRII/TGFBR2 and BMP receptor type II (BMPRII), respectively. The interaction between the ligand and the receptor induces the transphosphorylation of a receptor subunit, resulting in the activation of the SMAD-dependent (canonical) or non-SMAD-dependent signaling cascades, leading to the gene expression regulation and activation of the osteogenic transcription factors Distal-less Homeobox 5 (DLX5), RUNX2 and the zinc finger Osterix (OSX), also called SP7 (Figure 2) [44,57]. Finally, this results in the expression of specific osteoblast genes, including osteopontin (OPN), osteocalcin (OCN), osteonectin (ON) and collagen type 1 (COL1) and cell differentiation [58]. 

TGF-β and BMP signaling pathways are regulated by different components, which stimulate and/or inhibit osteoblast differentiation. They comprise BMP-binding proteins, such as Noggin, Chordin, Gremlin and Follistatin that sequester ligands, preventing binding to receptors, and intracellular inhibitory SMAD proteins, such as SMAD7, and the ubiquitin–proteasomal degradation pathway (Figure 2) [44,49].

The Wnt/β-catenin or canonical Wnt is another signaling pathway with an essential regulatory role in MSC self-renewal and differentiation [50]. The Wnt/β-catenin signaling pathway occurs in the early stage of osteogenesis, allowing the MSCs to differentiate into an osteoblast lineage whilst it is downregulated in differentiated cells [57]. Wnt signaling is initiated upon binding of Wnt ligands with the membrane receptor complex known as Frizzled/low-density lipoprotein receptor-related protein 5/6 (FZD/LRP) [59]. Interactions between Wnt ligands and FZD/LRP5/6 trigger disheveled (DSH) that switches off glycogen synthase kinase 3 β (GSK-3β) and enables β-catenin protein to be translocated into the nucleus, whereby it complexes with the binding T cell factor/lymphoid enhancer-binding factor (TCF/LEF) to stimulate the expression of target genes involved in osteogenesis, including *RUNX2* [60,61] (Figure 3). 

Several Wnt ligands, including Wnt4, Wnt5a, Wnt6, Wnt10a and Wnt10b, are involved in initiating the Wnt/β-catenin cascade and stimulation of osteoblast differentiation [62,63,64]. 

Contrariwise, Wnt3a and Wnt16 have negative effects on MSC osteogenic differentiation [65,66,67]. In addition, extracellular Wnt signaling antagonists can negatively modulate the Wnt/β-catenin signaling pathway, such as the secreted frizzled-related protein 1 and 2 (SFRP1 and SFRP2) [68], that bind Wnt ligands, preventing their interaction with the receptor [50], and the dickkopf-related protein1 (DKK1) [69], which binds the active site of the LRP5/6 receptors and allows proteasome β-catenin degradation through GSK-3β complex activation [70,71] (Figure 3). Another relevant Wnt signaling inhibitor is sclerostin (SOST) which binds the first beta propeller (E1) of the LRP5/6 Wnt co-receptors and mainly inhibits the Wnt1 canonical signaling pathway [50,72]. 

TGF-β/BMP and Wnt/β-catenin signaling pathways strictly interplay in a very complex way and affect each other in a positive feedback loop, allowing osteoblast differentiation in vitro and, thus, a proper skeletal development in vivo. Further, signals from the extracellular matrix (ECM) cooperate with osteogenic signaling pathways, mainly acting through integrins, the mediators of cell adhesion to the ECM and important mechano-transducers involved in both matrix deposition and organization [7,9,73]. Moreover, TGF-β/BMP and Wnt/β-catenin signaling pathways are components of a more complex molecular network with other different pathways and cytokines, such as Hedgehog (Hh), NOTCH, fibroblast growth factor (FGF) and parathyroid hormone-related peptide (PTHrP) that interplay for an appropriate osteoblast differentiation and bone development [44]. 

## 3. MiRNAs

MiRNAs are a class of non-coding RNAs (ncRNAs), generally 22 nucleotides long, with an essential role in gene regulation. They account for 1–5% of the human genome and regulate 30–60% of protein-coding genes [74]. MiRNAs are generated through a multi-step process (Figure 4). 

The gene silencing mechanism is determined by the degree and nature of complementarity between the miRNA seed site and the 3′-UTR of its target mRNA. When complementarity is complete, the mRNA target undergoes degradation, while, when it is partial, mRNA target protein levels are reduced [75]. Binding to mRNA represents the main activity of miRNA, although it is known that they can also regulate gene expression through other molecular mechanisms. MiRNAs can positively or negatively regulate osteogenic differentiation by targeting either osteogenic negative or positive regulatory genes and TFs, as shown below [76,77,78]. Due to these activities, miRNAs have been proposed as targets for innovative therapeutic approaches in several diseases, including bone-related diseases [79]. In tissue engineering approaches, miRNA mimics or antago-miRNAs are employed as bioactive factors [80] in combination with stem cells [5] or scaffolds [81] in order to improve bone tissue regeneration.

## 4. Osteogenic Differentiation by miRNA Regulation 

TGF-β/BMP and Wnt/β-catenin cascades are post-transcriptionally modulated by different miRNAs that stimulate or inhibit osteogenesis [45,50]. Specifically, several miRNAs can directly increase or reduce the expression of the genes coding for components of the signaling pathways and/or the TFs involved in osteogenic differentiation, ultimately exerting both stimulatory and inhibitory effects on osteogenesis, as reported herein (Table 1).

### 4.1. TGF-β/BMP Signaling Pathways

#### 4.1.1. TGF-β/BMP Ligands and Receptors

Few studies have reported on the miRNA regulation of TGF-β receptors in osteogenesis. Let-7a-5p targeted *TGFβRI* inhibiting the osteogenic differentiation of BM-MSCs in postmenopausal osteoporosis (PMOP) mice [82]. Several miRNAs target both BMP receptors and ligands. MiR-100 and miR-153 have been shown to attenuate human MSC (hMSC) osteogenic differentiation by targeting *BMP receptor type II* (*BMPR2*) [83,84]. Moreover, BMPR2 is a direct target of miR-155, whose inhibitory activity in osteogenic differentiation has been proven in C2C12 cells and *mouse embryonic fibroblasts* (*MEFs*) treated with bone morphogenetic protein 9 (BMP9) [85]. MiR-195-5p inhibited osteogenesis in periodontal MSCs (PDLSCs) from periodontitis patients by targeting *BMP receptor type IA* (*BMPR1A*) [86]. In the same study, it was also shown that miR-195-5p was regulated by mechanical loading and involved in mechanical loading-induced osteogenic differentiation. In hBM-MSCs, miR-125b binds to the 3′-UTR of the BMP receptor type 1B gene and inhibits differentiation [87]. Furthermore, it has been shown that miR-208a-3p targets the ACVR1/Alk2 gene inhibiting osteoblast differentiation in MC3T3-E1 cells and in vivo [88]. Regarding BMP ligands, miR-93-5p, miR-98 and miR-140-5p inhibit osteogenic differentiation of hMSCs by directly targeting BMP2 [89,90,91]. In BMP2-stimulated murine pre-osteoblast MC3T3-E1 cells, miR-370 attenuates osteogenic differentiation by targeting *BMP2* and the *Erythroblastosis virus E26 Oncogene Homolog 1* (*Ets1*) gene [92]. In hBM-MSCs, miR-214 reduces BMP2 expression binding to the 3′UTR and when overexpressed it inhibits osteogenic differentiation [93]. Analogously, direct interaction of miR-204 with BMP2 mRNA reduces differentiation of rat bone marrow MSCs [94]. 

#### 4.1.2. SMAD Cascade

Several miRNAs target the TGF-β signaling-induced SMAD2/3, the BMP signaling-induced SMAD1/5 and the TGF-β/BMP signaling-induced common SMAD4 affecting osteogenic differentiation. In TGF-β signaling SMADs, miR-10b promoted osteogenesis by targeting *SMAD2* in hASCs and enhanced bone formation in vivo, while its expression correlated with the bone formation *marker genes alkaline phosphatase* (*ALP*), *RUNX2* and *OPN* in clinical samples from patients affected by osteoporosis [95]. MiR-221-5p inhibited osteogenesis in myeloma bone disease mesenchymal stem cells (MBD-MSCs) by directly targeting *SMAD3*, which in turn negatively affected the PI3K/AKT/mTOR signaling pathway and, consequently, osteogenic differentiation [96]. *SMAD3* is also a direct target of miR-708, which suppressed osteogenesis and adipogenesis in MSCs, and it was found to be overexpressed in MSCs from patients with steroid-induced osteonecrosis of the femoral head (ONFH) [97]. 

In BMP signaling SMADs, miR-26a, a key player of skeletal muscle differentiation and regeneration [98], inhibited osteogenesis in hASCs by targeting *SMAD1*, while it activated osteogenic differentiation in Wnt signaling-induced BM-MSCs [99]. Members of the miR-30 family, such as miR-30a, -30b, -30c and -30d, inhibited osteogenesis in MC3T3-E1 by targeting *SMAD1* and *RUNX2*, and have been found to be downregulated during osteoblast differentiation [100]. Further, miR-222-3p appears as a negative regulator of the osteogenic differentiation, as its overexpression significantly suppressed *SMAD5* and *RUNX2* protein levels, whereas its inhibition increased their expression in human BM-MSCs [101]. Similarly, miR-133 and miR-135 suppressed osteogenesis targeting *RUNX2* and *SMAD1/5*, respectively, in C2C12 mouse mesenchymal progenitors [102]. In agreement, the downregulation of miR-133a, -133b and -135a found in human dental pulp stem cells (DPSCs) grown on titanium disks allowed the target genes *RUNX2* and *SMAD5* to be expressed, leading to osteogenic differentiation [103]. Tang et al. reported that miR-203-3p targets the 3′-UTR of *SMAD1* mRNA inhibiting in vitro and in vivo osteogenesis in diabetic rats, suggesting miR-203-3p as a potential therapeutic target in diabetic bones for ameliorating osteoporosis and fracture healing [104].

Finally, in TGF-β/BMP signaling-induced common SMADs, miR-144-3p negatively regulated osteogenic differentiation and proliferation in embryo cells from C3H mice (C3H10T1/2) cells by directly targeting *SMAD4* [105]. Similarly, miR-146a negatively regulated osteogenesis and bone regeneration through interaction with the 3′ untranslated region (3′-UTR) of *SMAD4* mRNA, both in vitro and in vivo [106]. 

#### 4.1.3. TGF-β/BMP Signaling Pathway Intracellular and Extracellular Inhibitors: SMAD7, HDAC4, BMP Antagonist Proteins

The *SMAD7* gene has an antagonistic role in TGF-β/BMP signaling [107] and appears to be regulated by several miRNAs [108,109,110]. MiR-17-5p, which is part of the miR-17/92 cluster and has important roles in several cancers, directly targets *SMAD7* in human BM-MSCs, increasing their osteoblastic differentiation and cell proliferation [111]. MiR-21, largely known for its regulatory role in bone [112], induces BM-MSC bone regeneration through the SMAD7-SMAD1/5/8-RUNX2 pathway [107]. The link between miR-21a and *SMAD7* was confirmed by a very recent study, demonstrating that phytol, derived from aromatic plants, is able to promote osteoblast differentiation in C3H10T1/2 cells by downregulating *SMAD7* through miR-21a, thus leading to the increased expression of RUNX2 [113]. Similarly, the expression of miR-21-5p was also increased and associated with a reduced expression of *SMAD7* during the pulsed electromagnetic field (PEMF)-induced osteogenic differentiation of human BM-MSCs, stimulating the TGF-β signaling pathway [114]. Further, in a recent study, high levels of miR-21-5p, as well as of the pro-osteogenic miR-129-5p and miR-378-5p, have been identified in hMSCs differentiated by treatment with serum from runners after a half marathon. This suggests that miRNAs can also regulate MSC osteogenic differentiation in response to physical exercise [115]. By targeting *SMAD7*, miR-590-5p indirectly protects and stabilizes RUNX2 in C3H10T1/2 and MG63 cells [116]. Interestingly, MC3T3-E1 cells exposed to elevated glucose conditions showed increased *SMAD7* levels with significant downregulation of miR-590-5p and osteoblastic proteins, e.g., collagen I, RUNX2 and ALP, suggesting that miR-590-5p could be a potential target for treatment for diabetic osteoporosis [117]. 

Several extracellular proteins act as antagonists of BMP signaling. They include Gremlin, noggin and Follistatin [118]. To the best of our knowledge, only miR-27a, targeting Gremlin 1 (GREM1) has been involved in the regulation of osteogenic and adipogenic differentiation in rat BM-MSCs under steroid treatment [119].

HDAC is a family of key epigenetic factors that regulates the expression of genes in concert with DNA methylation and miRNA expression [120,121,122,123,124]. In the TGF-β signaling pathway, the HDAC4/5 complex binds the nuclear phosphorylated R-SMAD SMAD2/3 to repress *RUNX2* expression, resulting in osteoblast differentiation inhibition [125]. The overexpression of miR-29a and miR-29b blocked *HDAC4*, thus inducing osteoblast differentiation in primary BM-MSCs and MC3TC-E1 cells [126,127]. Due to miR-29b activity, the R9-LK15/miR-29b nano-complex that promotes osteogenic differentiation with high transfection efficiency has recently been developed [45]. Further, the miR-29b level has been increased following exposure of hBMSCs to external stimuli such as PEMFs [128]. 

### 4.2. Wnt/β-Catenin Signaling Pathway

#### 4.2.1. Wnt Ligands and Receptors

During osteoblast differentiation, the expression of miR-16-2-3p, also named miR-16-2*, is significantly decreased in human BM-MSCs, whereas its overexpression impaired osteogenic differentiation blocking the Wnt signal pathway by directly targeting *Wnt5a* mRNA [129]. In another study, Wang et al. reported that *Wnt5a* mRNA can also be a target of miR-1297, which was found to be highly expressed in sera of osteoporotic patients, and significantly decreased in BM-MSCs after osteogenic induction [130]. Thus, the increased expression of miR-1297 may participate in osteoporosis progression by targeting *Wnt5a*. In this context, miR-9-5p also promotes the occurrence and progression of osteoporosis through inhibiting osteogenesis and promoting adipogenesis via targeting *Wnt3a* in MSCs [131]. *Wnt5a* and FZD3 expression has also recently been found to be inhibited by miR-38, and binding to the mRNA 3′UTR hampered osteogenic differentiation of BM-MSCs [132]. The same authors also reported the involvement of miR-139-5p in hBM-MSC osteogenesis by directly targeting the *β-catenin* gene and frizzled 4 (FZD4) [133]. In addition, low-density lipoprotein receptor-related protein 6 (LRP6) and low-density lipoprotein receptor-associated protein 5 (LRP5), critical co-receptors for Wnts, have been shown to be direct targets of miR-30e and miR-23a, respectively. It was described that miR-30e reciprocally regulates the differentiation of adipocytes and osteoblasts by directly targeting LRP6 [134] and miR-23a overexpression inhibited osteogenic differentiation of hBM-MSCs [135].

Other Wnt ligands have been identified as targets of specific miRNAs. Wnt6 and Wnt10a were identified as targets of miR-378. In this study, the use of miR-378 mimics could suppress osteogenesis in hMSCs, whereas anti-miR-378 allowed their osteogenic differentiation, thus confirming that miR-378 inhibited osteogenesis via inactivating Wnt/β-catenin signaling [136]. Further, miR-154-5p negatively regulates hASC osteogenic differentiation under tensile stress, through the Wnt/PCP pathway by directly targeting *Wnt11* [137].

#### 4.2.2. Transcription Factors: β-Catenin and TCF/LEF

β-catenin is a fundamental factor in the Wnt/β-catenin signaling pathway which binds to TCF/LEF transcription factors, leading to the regulation of genes involved in osteogenic differentiation [138]. Few miRNAs have been found to be involved in osteogenic differentiation though binding to β-catenin and TCF/LEF mRNAs, although several miRNAs targeting these genes have been reported in cancer [139]. MiR-132 inhibited the osteogenic differentiation of umbilical cord MSCs (UC-MSCs) targeting β-catenin and resulting in the downregulation OSX expression [140]. By targeting TCF-1, miR-24 overexpression decreased the expression of osteogenic differentiation markers [141]. MiR-129-5p targets TCF-4 and its downregulation enhanced osteoblast differentiation in the MC3T3-E1 cell line, and in C57BL6 mice it ameliorated osteoporosis [142]. Further, it has been shown that miR-26a and miR-26b, targeting glycogen synthase kinase three beta (GSK3β), lead to Wnt signaling activation and promote osteogenic differentiation of BM-MSCs [99,143]. Interestingly, miR-26a is increased during osteogenic differentiation induced by PEMFs [128]. 

#### 4.2.3. Wnt Cascade Inhibitors

As mentioned above, APC, GSK-3 β, CK1α and Axin form the β-catenin destruction complex, which operates by modulating β-catenin levels. MiR-27a and miR-142-3p downregulated APC and therefore activated osteoblastic differentiation in human fetal osteoblastic cell line 1.19 (hFOB1.19) through the Wnt signaling pathway [144,145]. In hMSCs, miR-590-3p bound to the 3′-UTR of *APC* mRNA, leading to osteogenic differentiation via β-catenin stabilization [44]. GSK3β, an osteogenesis suppressor, participates in the silencing of the Wnt signaling pathway by phosphorylating cytoplasmic β-catenin. MiR-26a, miR-346 and miR-199b-5p directly target the 3′-UTR of *GSK3β* mRNA and prevent its translation. In agreement with their role, these miRNAs are maintained at high expression levels during osteoblastic differentiation of mouse and human BM-MSCs [99,146,147]. 

Among the several extracellular Wnt signaling antagonists, DKK1, highly expressed in osteoblasts and MSCs, is a pivotal inhibitor of the Wnt cascade [50]. MiR-9, miR335-5p, miR-433-3p and miR-217 promoted osteoblastic differentiation in mouse and human MSCs targeting the 3′-UTR region of *DKK1* mRNA [148,149,150,151]. 

Of note, miR-335-5p was significantly upregulated in bone marrow stem cells during low-magnitude, high-frequency vibration-induced osteogenic differentiation [152].

Additionally, miR-146a inhibited *DKK1* expression by directly targeting its 3′-UTR mRNA, and was found to be upregulated in hip capsule tissues from patients affected by ankylosing spondylitis (AS) [153]. MiR-218 has shown stimulatory effects on the Wnt signaling pathway, through the downregulation of different inhibitors, including *DKK2*, *SFRP2* and *sclerostin* (*SOST*) in mice BM-MSCs, MC3T3 and hASCs [154,155]. Further, miR-96 binds to the SOST gene and inhibits its expression. In agreement with this, the overexpression of miR-96 increases osteoblast differentiation and bone formation in AS-affected mice [156]. During osteoblastic differentiation of hFOBs, miR-27a was upregulated and targeted the extracellular antagonist of Wnt signaling pathway *SFRP1* [157]. Thus, miR-27a has dual beneficial effects on the Wnt signaling pathway by positively regulating it through the downregulation of *APC* and *SFRP1* [144,157]. 

#### 4.2.4. Transcription Factor RUNX2 

CBFA-1/RUNX2 is the master transcription factor for osteogenic differentiation [158]. As reported above, its expression is regulated by several signaling pathways, especially by BMP and Wnt [159]. In addition, RUNX2 can regulate TCF/LEF family gene expression, and the β-catenin-TCF/LEF dimer can bind to the RUNX2 promoter and promote its transcriptional activity. Because of the key role of RUNX2 in the differentiation of MSCs into osteoblasts [160], several studies have been focused on RUNX2 regulation by miRNAs and recently reviewed [159,161].

Zhang et al. found a panel of 11 RUNX2-targeting miRNAs, such as miR-23a, miR-30c, miR-34c, miR-133a, miR-135a, miR-137, miR-204, miR-205, miR-217, miR-218 and miR-338, expressed in a lineage-related pattern in different mesenchymal cell types. Apart from miR-218, all RUNX2-targeting miRNAs inhibited osteoblast differentiation in the osteoblastic cell line MC3T3 [162]. Conversely, miR-218 downregulated RUNX2 expression in undifferentiated human-derived DPSCs [163]. Moreover, computational analyses have been used to identify miR-23a, miR-23b, miR-30b, miR-143, miR-203, miR-217 and miR-221 in the potential direct regulation of RUNX2 during the differentiation of MSCs to pre-osteoblasts [164]. 

Most of the miRNAs reported above have been investigated in different studies, confirming or not their role in osteogenic differentiation [135,151,165,166,167,168]. Great interest was shown for miR-23a, which was found to be overexpressed in mouse non-transformed osteoblastic cell line MC3T3-E1, inhibiting osteogenic differentiation [162,169]. In PDLSCs, miR-23a prevented osteogenic differentiation, and its expression was suggested to be a potential biomarker of periodontitis [170]. Differently, Park and collaborators reported the limited role of miR-23a in osteogenesis and bone homeostasis in vivo [171]. The activity of miR-30 in the downregulation of RUNX2 by binding its mRNA 3′-UTR sequence influencing osteogenic differentiation has been confirmed in the mesenchymal stem cell line C3H10T1/2 [172] and more recently in hASCs [173]. Additionally, the role of miR-133, miR-133a, miRNA-133a-5p and miR-135a as miRNAs targeting RUNX2 and inhibitors of osteogenesis has been investigated and confirmed in different osteoblastic cell lines, including C2C12 cells [162,174] and MC3T3-E1 cells [162,175]. Notably, miR-135a-5p was overexpressed in osteoporotic post-menopausal females when analyzing the RNA extracted from the bone tissue fragments of postmenopausal women with and without osteoporosis [176]. Other miRNAs have been found to be involved in the regulatory mechanisms of osteogenesis in osteoporosis. MiR-338-3p expression was found to be downregulated during the osteoblastic differentiation of BM-MSCs, and increased in BM-MSCs derived from osteoporotic mice, whereby miR-338-3p inhibited osteogenic differentiation by targeting RUNX2 and fibroblast growth factor receptor 2 (FGFR2) [177]. MiR-205 has been found to be involved in altered osteogenic differentiation of human aortic smooth muscle cells (HASMCs) [178]. Further, it inhibits osteogenic differentiation in a female mouse model of type 2 diabetes mellitus (T2DM) and osteoporosis (OP) and, in agreement with this, high levels of miR-205 have been identified in human female patients. This shows the correlation between the in vitro and in vivo activities of this miRNA [179], which negatively regulates osteoblastic differentiation of BM-MSCs, via inhibition of RUNX2 as well as special AT-rich sequence-binding protein 2 (SATB2) expression [180]. 

MiRNAs have been investigated in other bone disorders or the alteration of osteogenic differentiation. Recently, miR-137-3p, directly targeting RUNX2, has been found to be involved in osteonecrosis of the femoral head. It also showed the ability to target CXCL12, coding for SDF-1α, an angiogenic factor and, notably, the silencing of miR-137-3p could favor both osteogenesis and angiogenesis [181]. MiR-204 deficiency has been found to be involved in the increased osteogenic activity associated with calcific aortic valve disease [182]. This result agrees with several studies reporting that in different mesenchymal progenitor cell lines, such as C3H10T1/2, C2C12, ST2 and primary BM-MSCs, miR-204, targeting RUNX2, negatively controls osteogenic differentiation [94,183]. The findings of a recent study provided evidence that miR-628-3p is upregulated in patients with atrophic non-union, a serious complication of fractures, and may exert an inhibitory effect on osteogenesis by suppressing its target gene RUNX2. Indeed, in vitro experiments on human osteoblast-like cells (MG63) transfected with miR-628-3p mimics led to decreased osteogenic differentiation [184].

Different miRNAs can cooperatively regulate a common signaling pathway by affecting one specific transcription factor complex. For instance, miR-34c and miR-145 cooperate to inhibit osteoblast differentiation of MC3T3-E1 cells by targeting RUNX2 and core binding factor beta (CBFB), respectively, preventing the transcription factor complex RUNX2/CBFB from being formed and subsequent transcription of RUNX target genes [185]. Recently, miR-505 was identified as a new regulator of RUNX2 [186]. Specifically, in MC3T3-E1 cells, miR-505 was downregulated during osteogenic differentiation in vitro, whilst its overexpression has been found to inhibit osteogenic differentiation and suppress osteoblast cell growth. Therefore, miR-505 may be a new potential therapeutic target for promoting new bone regeneration [186].

## 5. Conclusions

MSCs are the key players in bone formation and natural bone repair processes as well as in bone tissue engineering approaches. Dysregulation of MSC activities is involved in several bone disorders, including osteoporosis [187]. Growing evidence shows that MSC differentiation is regulated by different epigenetic factors, including miRNAs. Since the discovery of miRNAs, there have been great advancements in miRNA knowledge by both mechanistic studies and studies evaluating miRNAs as diagnostic and predictive biomarkers and therapeutics. The present review highlighted miRNAs as powerful post-transcriptional regulatory factors of the osteogenic process and was focused on the miRNAs which directly target genes coding for components of the main signaling pathways involved in osteogenesis. Each signaling pathway includes extracellular ligands, membrane receptors and intracellular proteins which act as mediators of the pathway. Indeed, data show that miRNAs positively or negatively regulate osteogenic differentiation by targeting genes involved in TGF-β/BMP and Wnt/β-catenin signaling pathways. A number of miRNAs suppress osteogenesis by directly targeting mRNAs from genes coding for positive osteogenic factors, such as: (i) the extracellular ligands *TGF-β, BMP* and *Wnt*; (ii) the TFs *RUNX2* and *TCF-1*. On the other hand, miRNAs can promote osteogenesis by directly targeting mRNAs from genes coding for negative osteogenic factors, such as: (i) *HDAC4*, *SMAD7*, *DKK1*, *GSK-3β*, *APC*, *SFRP1* and *SFRP2* together with *DKK2* and *SOST*. Interestingly, some miRNAs display both stimulatory and inhibitory effects during osteogenesis. As each miRNA can target different genes, these data indicate that the activity of miRNA can differ in different MSCs, which may be characterized by specific regulatory signaling pathways and molecules, including other non-coding RNAs [99,188]. 

Interestingly, some studies have begun to show that miRNAs associated with the main osteogenic pathways can be modulated by physical stimuli, including electromagnetic field stimulation and mechanical forces [128]. In this context, it has been recently reported that physical exercise can induce the expression of pro-osteogenic miRNAs, indicating a molecular basis for environmental effects on bone [115]. At the cellular level, the integrins are important mechano-transducers involved in both matrix deposition and organization. In spite of their relevant role as modulators of TGF-β/BMP and Wnt/β-catenin signaling pathways, to date, few miRNAs targeting integrins have been identified in osteogenic differentiation [189]. Further studies identifying the biological interaction between integrins and miRNAs in osteogenesis are needed [189].

Notably, recent studies have begun to identify altered levels of specific miRNAs which target genes of the BMP or Wnt pathways in bone disorders, including osteoporosis or osteonecrosis [97,176,190]. Because of the essential roles of these pathways in the osteogenic process, these results are expected, but, on the other hand, they may be helpful in planning therapeutic approaches. 

In conclusion, a lot of evidence supports miRNA involvement in the osteogenic process and bone tissue formation, although the related mechanisms remain poorly understood. Understanding the mechanisms of osteoblast differentiation regulated by specific miRNAs, as well as stimuli able to regulate their expression, will be significant to develop new therapeutics for the treatment of bone disorders, and guide lifestyles. Further studies aiming to discover novel miRNAs and to understand their complex molecular activities on their gene targets are needed. In this context, knowledge on the tissue-specific MSC biology and a deep understanding of the complex relationships among miRNAs, other classes of non-coding RNAs and their target genes are required. Further, crossing data concerning altered miRNAs in specific pathological conditions with their experimental established functional activity may also direct the clinical approaches based on the use miRNA mimics or antago-miRNAs in bone disorders. 

## Figures and Tables

**Figure 1 ijms-22-02362-f001:**
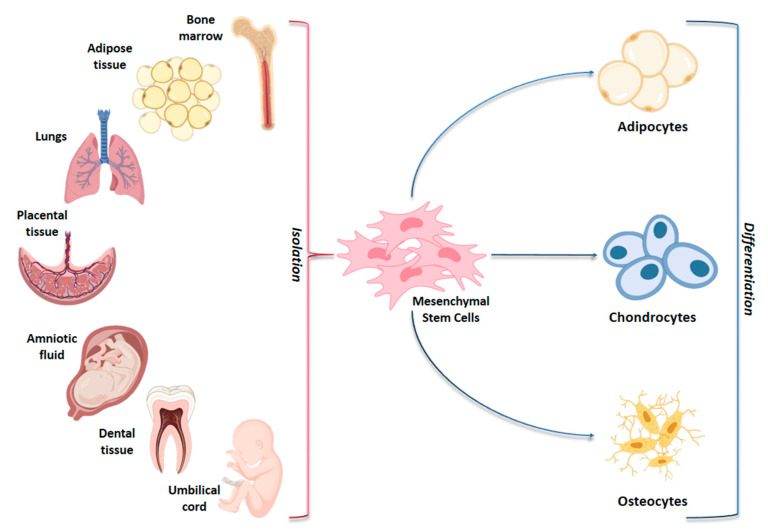
Schematic representation of adult and fetal/neonatal tissue sources of mesenchymal stem cells (MSCs) and their potential of differentiation in various cell lines. MSCs can be isolated from several tissue sources in the body and they may differentiate into adipocytes, chondrocytes and osteocytes.

**Figure 2 ijms-22-02362-f002:**
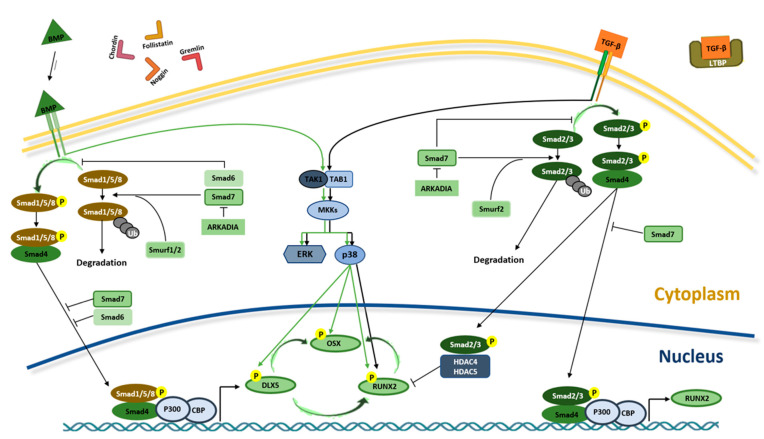
BMPs and TGF-β signaling pathways in osteogenesis. Interaction between TGF-βs and BMPs and receptors initiate SMAD-dependent and non-SMAD-dependent signaling pathways. In SMAD-dependent signaling pathway, interaction between TGF-β and receptor leads to SMAD2/3 (R-SMAD) phosphorylation. Phosphorylated R-SMAD (SMAD2/3-Pi) binds SMAD4 protein and migrates to the nucleus of the cell where, alongside P300 and CREB-binding protein (CBP) coactivators, it controls *RUNX2* expression. In the nucleus, SMAD2/3-Pi without SMAD4 recruits HDAC4 and HDAC5 and blocks RUNX2 activity. In BMP signaling, R-SMAD (SMAD1/5/8) binds SMAD4 and migrates to the nucleus, inducing *RUNX2* and *OSX* expression through *DLX5* activation. In the non-SMAD-dependent signaling pathway, TAK1 and TAB1 activate the MKKs p38 MAPK or Erk cascades, inducing DLX5, RUNX2 and OSX phosphorylation. The interaction between TGF-β and its receptor is blocked by the latent TGF-β binding protein (LTBP), which binds to TGF-β. Similarly, in order to prevent the BMP–receptor interaction, Gremlin, Chordin, Follistatin and Noggin proteins bind to BMP. The translocation of SMAD2/3 and SMAD1/5/8 into the nucleus is prevented by SMAD7 that, along with SMURF1/2, induce their proteasome-mediated degradation. In addition, the binding between ARKADIA and SMAD6/7 proteins positively regulates osteoblast differentiation.

**Figure 3 ijms-22-02362-f003:**
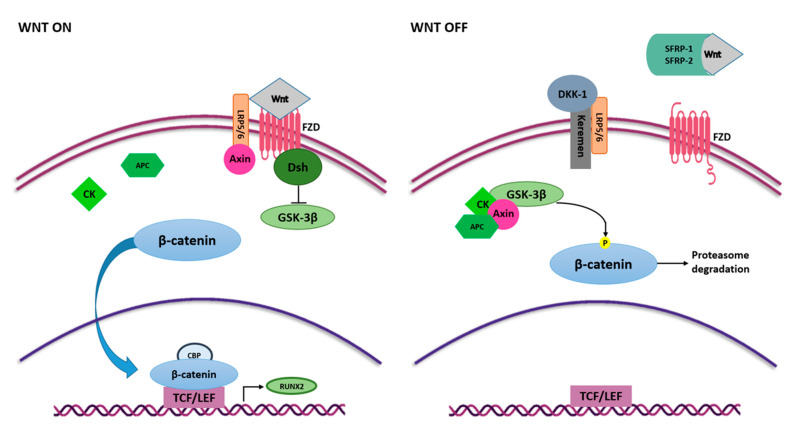
Wnt/β-catenin signaling pathway in osteogenesis. Wnt ligands interact with FZD and activate Wnt signaling pathway. FZD binds Dsh that inhibits GSK3-β activity, and thus the phosphorylation of β-catenin. β-catenin translocates into the nucleus, where it binds to TCF/LEF and CBP and induces the expression of *RUNX2*. Interaction between Wnt ligands and SFRP-1/2 and the binding of LRP5/6 and Keremen with DKK-1 antagonist switch off Wnt signaling pathway. In this case, GSK3-β, together with Axin, CK and APC, phosphorylate β-catenin, inducing proteasome-mediated degradation.

**Figure 4 ijms-22-02362-f004:**
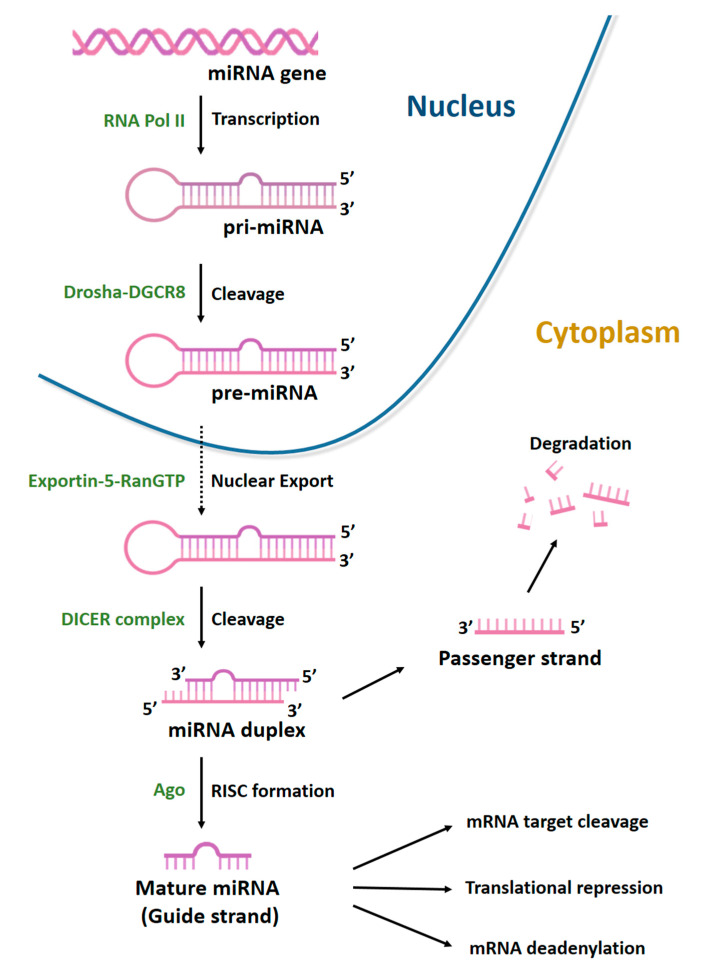
Biogenesis of microRNAs. MiRNAs are transcribed by RNA polymerase II (Pol II) to generate a long precursor transcript named primary microRNA (pri-miRNA). This RNA molecule folds up into a secondary structure (stem-loop) to form a partial double helix, composed of 100–1000 nt with a 5′-cap. MiRNA maturation process can be divided into three phases. In the first, known as cropping, pri-miRNA is converted into the precursor miRNA (pre-miRNA) via the cutting activity of the Drosha enzyme, a nuclear endoribonuclease III. Pre-miRNA has a hairpin structure (stem-loop) and a length of about 70–80 nt. Following cropping, the pre-miRNA has a 5′P and a 3′OH and 2–3 nt at the protruding end with a single strand. In the second phase, the nuclear export factors Exportin-5 and ras-related nuclear protein (RAN-GTP) mediate the export of pre-miRNAs from nucleus to cytoplasm. In the third phase, known as dicing, another type of RNA endonuclease III, known as Dicer, processes pre-miRNA in the cytoplasm by cleaving it into an 18–22 nt double-stranded miRNA (miRNA duplex). Lastly, mature miRNA incorporated into the RNA-induced silencing complex (RISC) is able to bind the 3′UTR region of its target mRNA.

**Table 1 ijms-22-02362-t001:** MicroRNAs, their direct target genes and cell types in which they play a role during osteogenesis.

miRNA	Cell Type	Target Gene	Effect on Osteogenesis
let-7a-5p	PMOP BM-MSCs	TGFβR1	−
miR-9	MSCs	DKK1	+
miR-9-5p	MSCs	Wnt3a	−
miR-10b	hASCs	SMAD2	+
miR-16-2-3p	hBM-MSCs	Wnt5a	−
miR-17-5p	hBM-MSCs	SMAD7	+
miR-21	BM-MSCs	SMAD7	+
miR-21-5p	BM-MSCs	SMAD7	+
miR-21a	C3H10T1/2	SMAD7	+
miR-23a	MSCs, PDLSCs, MC3T3-E1, hBM-MSCs	RUNX2, LRP6	−
miR-23b	MSCs	RUNX2	−
miR-24	BM-MSCs, MC3T3-E1	TCF-1	−
miR-26a	BM-MSCs, hASCs	SMAD1, GSK3β	−/+
miR-26b	BM-MSCs	GSK3β	+
miR-27a	hFOB1.19, BM-MSCs	APC, SFRP1, GREM1	−/+
miR-29a	BM-MSCs, MC3TC-E1	HDAC4	+
miR-29b	BM-MSCs, MC3TC-E1	HDAC4	+
miR-30	MC3T3-E1, MSCs, hASCs, C3H10T1/2	RUNX2, SMAD1	−
miR-30a	MC3T3-E1	RUNX2, SMAD1	−
miR-30b	MSCs, MC3T3-E1	RUNX2, SMAD1	−
miR-30c	MSCs, hASCs, MC3T3-E1	RUNX2, SMAD1	−
miR-30d	MC3T3-E1	RUNX2, SMAD1	−
miR-30e	hBM-MSCs	LRP6	−
miR-34c	MSCs, MC3T3-E1	RUNX2	−
miR-93-5p	MSCs, BM-MSCs	BMP2	−
miR-96	Mice AS cells	SOST	+
miR-98	hMSCs	BMP2	−
miR-100	hMSCs	BMPR2	−
miR-125b	hBM-MSCs	BMPR1B	−
miR-129-5p	MC3T3-E1, C57BL6	TCF-4	−
miR-132	UC-MSCs	β-catenin	−
miR-133	C2C12	RUNX2	−
miR-133a	DPSCs, C2C12	RUNX2, SMAD5	−
miR-133a-5p	MC3T3-E1	RUNX2	−
miR-133b	DPSCs	RUNX2, SMAD5	−
miR-135	C2C12	SMAD1, SMAD5	−
miR-135a	MSCs, MC3T3-E1, ATDC5, C2C12, DPSCs	RUNX2, SMAD5	−
miR-135a-5p	C2C12	RUNX2	−
miR-137	MSCs	RUNX2	−
miR-137-3p	BM-MSCs	RUNX2	−
miR-139-5p	hBM-MSCs	β-catenin, FZD4	−
miR-140-5p	hMSCs	BMP2	−
miR-142-3p	hFOB1.19	APC	+
miR-143	MSCs	RUNX2	−
miR-144-3p	C3H10T1/2	SMAD4	−
miR-145	MC3T3-E1	CBFB	−
miR-146a	hASCs, AS fibroblast	SMAD4, DKK1	−/+
miR-153	hMSCs	BMPR2	−
miR-154-5p	hASCs	Wnt11	−
miR-155	C2C12, MEF	BMPR2	−
miR-195-5p	PDLSCs	BMPR1A	−
miR-199b-5p	hBM-MSCs	GSK3β	+
miR-203	MSCs	RUNX2	−
miR-203-3p	MSCs	SMAD1	−
miR-204	C3H10T1/2, C2C12, ST2, BM-MSCs	RUNX2, BMP2	−
miR-205	MSCs, BM-MSCs	RUNX2, SATB2	−
miR-208a-3p	MC3T3-E1	ACVR1/Alk2	−
miR-214	BM-MSCs	BMP2	−
miR-217	MSCs	RUNX2, DKK1	−/+
miR-218	hASCs, BM-MSCs, MC3T3, hDPSCs	DKK2, SFRP2, SOST, RUNX2	+/−
miR-221	MSCs	RUNX2	−
miR-221-5p	MB D-MSCs	SMAD3	−
miR-222-3p	BM-MSCs	RUNX2, SMAD5	−
miR-335-5p	MSCs, C3H10T-1/2, MC3T3-E1, MLO-A5, -Y4	DKK1	+
miR-338	MSCs	RUNX2	−
miR-338-3p	BM-MSCs mice	RUNX2, FGFR2	−
miR-346	hBM-MSCs	GSK3β	+
miR-370	MC3T3-E1	BMP2	−
miR-378	MSCs	Wnt6, Wnt10a	−
miR-381	hBM-MSCs	Wnt5a, FZD3	−
miR-433-3p	MSCs, hFOB1.19, ROS17/2.8	DKK1	+
miR-505	MC3T3-E1	RUNX2	−
miR-590-3p	hMSCs	APC	+
miR-590-5p	C3H10T1/2, MG63, MC3T3-E1	SMAD7	+
miR-628-3p	MG63	RUNX2	−
miR-708	MSCs	SMAD3	−
miR-1297	hBM-MSCs	Wnt5a	−

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
