# Peer review of "MicroRNAs Modulate Signaling Pathways in Osteogenic Differentiation of Mesenchymal Stem Cells"

_ijms, 2021, doi:10.3390/ijms22052362_

Round 1

Reviewer 1 Report

The review is not particularly interesting and reports well-known concepts. The structure is not well balanced and needs to be revised. Paragraph 2 should be shortened as it takes the reader away from the topic of the review. Paragraph 3 reports well-known concepts and should be shortened. It would also be appropriate, to enrich the content of the review, to discuss the role of paraphysiological stimuli, such as physical exercise, in modulating the expression of microRNAs in osteogenic differentiation.

Author Response

PLEAS SEE THE ATTACHEMENT

Reviewer 2 Report

I would like to congratulate the authors for this comprehensive review of the role of miRNAs in osteogenic differentiation of MSCs. A few comments would allow publication of the review:

The running title is not representative since you are not talking about osteogenesis in general. please be more specific.

Interesting presentation of MSC bone signalling pathways. However there is no mention about the importance of integrin signalling in MSC osteogenesis. There are ways that integrin signalling is exploited in osteogenesis of MSCs eg.  Di Benedetto et al. Stem Cell Res 2015 --- Klontzas et al. Acta Biomater 2019. Therefore it needs to be mentioned.

Please check if there is literature on miRNAs and integrins in osteogenesis.

Please include a paragraph where you talk about other ways of epigenetic regulation in osteogenesis.

Round 2

Reviewer 1 Report

The authors responded correctly to the requested review and the paper is significantly improved.